# Modeling determinants of time-to-premarital cohabitation among Ethiopian women using parametric shared frailty models

Woldemariam Erkalo Gobena[1]*, Wubishet Gezimu[2], Gizachew Gobebo Mekebo[3], Teramaj Wongel Wotale[4], Mesfin Esayas Lelisho[5]

1 Department of Statistics, College of Natural Science, Mattu University, Mettu, Ethiopia, 2 Department of Nursing, College of Health Sciences, Mattu University, Mettu, Ethiopia, 3 Department of Statistics, College of Natural Science, Ambo University, Ambo, Ethiopia, 4 Department of Statistics, College of Natural Science, Dilla University, Dilla, Ethiopia, 5 Department of Statistics, College of Natural Science, Mizan-Tepi University, Tepi, Ethiopia

* woldenegniko@gmail.com

## Abstract

**Data Availability Statement:** Third-party data was obtained for this study from the DHS Program. Data may be requested from the DHS Program

### Background

Premarital cohabitation is rampant and currently practiced worldwide, particularly in sub-Saharan Africa. It is a known cause of marital instability and divorce. It is also associated with intimate partner violence and harms the psychology of children in later life. However, in Ethiopia, there has been limited attention given to premarital cohabitation.

### Objective

The main goal of this study was to identify the determinants of time-to-premarital cohabitation among Ethiopian women.

### Methods

The 2016 EDHS data was used to achieve the study's goal. The survival information of 15683 women was analyzed based on their age at premarital cohabitation. The regional states of the women were used as a clustering effect in the models. Exponential, Weibull, and Log-logistic baseline models were used to identify factors associated with age at premarital cohabitation utilizing socioeconomic and demographic characteristics.

### Results

The median age of premarital cohabitation was found to be 18 years. Surprisingly, 72.7% of participants were cohabitated in the study area. According to the Log-logistic-Gamma shared frailty model, place of residence, occupation, educational status, and being pregnant were found to be factors determining the time to premarital cohabitation.

after creating an account. More access information can be found on the DHS Program website (https://www.dhsprogram.com/). The authors confirm that interested researchers would be able to access these data in the same manner as the authors. The authors also confirm that they had no special access privileges that others would not have. The preprint of this manuscript can be accessed at https://assets.researchsquare.com/files/rs-291159/v1_covered.pdf?c=1631859250.

**Funding:** The author(s) received no specific funding for this work.

**Competing interests:** The authors have declared that no competing interests exist.

**Abbreviations:** AIC, Akaike's Criterion; CSA, Central Statistical Agency; EDHS, Ethiopian Demographic and Health Survey; SSA, sub-Saharan Africa; USA, United States of America.

## Conclusion

Premarital cohabitation among Ethiopian women was higher compared to women in the sub-Saharan Africa and East Africa. Place of residence, occupation, educational status, and being pregnant were found to be factors determining the time for premarital cohabitation. Therefore, we recommend the concerned bodies set out strategies to educate women about the influencing factors and dangers of premarital cohabitation.

## Introduction

Premarital cohabitation is defined as a woman and man living together and having a sexual relationship without being formally married [1]. Some identified causes of premarital cohabitation include financial incapability, poor religious and moral values, and couples' intention to test life [2].

In recent decades, premarital cohabitation has become more common around the world [3]. It has been extensively examined from a demographic standpoint in various parts of the world, such as North America, Latin America, Western Europe, and Eastern Europe, where it has been observed to function as an alternative or stepping stone to marriage [4]. In the United States of America (USA), about 70% of couples cohabited before their marriage [5]. Cohabitation is also very common in sub-Saharan Africa (SSA) [4]. A study conducted in Nigeria found 65.2% of premarital cohabitation [6]. In East Africa, about 44% of young women were cohabiting [7]. Premarital cohabitation was frowned upon in Ethiopia and was regarded as a socially unacceptable practice [8].

When compared to married couples, cohabitated couples have a higher probability of divorce and exhibit inferior interpersonal skills and less predictable attitudes toward marriage [9,10]. Cohabitation may expose the couples to gender-based violence, particularly intimate partner violence [2,11]. As a result, cohabitation may alter an individual's perceptions of marriage, making them less dedicated and devoted to the institution [12]. Evidence shows that premarital cohabitation can decrease matrimonial satisfaction, increase unfaithfulness, and may cause dissolution or divorce among couples [13–16]. Premarital cohabitation can also result in reproductive health problems such as unwanted pregnancy and sexually transmitted diseases (STDs) [2]. In addition, marriage after premarital cohabitation can detriment the mental health of the couple [17]. Besides its effect on the couple's life, premarital cohabitation can cause various psychological, social, and economic traumas for their future children [15,18].

The socio-demographic and economic status of individuals is the known influencer of premarital cohabitation. According to a study conducted in the USA, lower education, weak religious values, and poor backgrounds were found to be inducing factors in premarital cohabitation [5]. Similarly, socio-demographic factors, including age, level of education, religion, wealth index, and residence place, showed a significant association with premarital cohabitation, according to a systematic review conducted in SSA [4]. Another demographic study conducted in East Africa identified age, religion, educational status, occupational status, wealth index, and Ugandan women as determinants of premarital cohabitation [7]. Economic insufficiencies such as lack of shelter (accommodation) and poverty are also determining premarital cohabitation [19].

Despite its rising nature and noted negative life outcomes among couples and their children, premarital cohabitation gets little attention from the scientific community, particularly in Ethiopia. Hence, the current study was intended to fill the literature gap by investigating the determinants of time-to-premarital cohabitation among Ethiopian women using the

parametric shared frailty models. The findings of this study can benefit individuals and policy-makers in understanding the influencers of premarital cohabitation.

## Materials and methods

### Data source

The information for this study came from the Ethiopian Demographic and Health Survey data (EDHS-2016). For this study, the 2016 EDHS sample was selected using a stratified, two-stage cluster sampling design, and house-to-house cross-sectional survey data extracted from the 2016 EDHS were used [20]. Since this study was observational, we have used a "Strengthening the reporting of observational studies in epidemiology (StroBE) statement: guidelines for reporting observational studies." to report the methodical flows and findings (S1 File).

### Study population and variables

**Study population.**   In the families of selected clusters (regions), a total of 15,683 women between the ages of 10 and 43 were identified. As a result, the risk variables for women's pre-marital cohabitation were examined in this study, which included 15,683 women.

**Study variables. Outcome variable**:—The time to premarital cohabitation among Ethiopian women, assessed in years, is the response variable in this study. It is calculated as the time between birth and the age of first cohabitation. For this study, cohabited women were an event of interest, and women who did not participate in the event (not cohabited) were censored.

**Independent factors**:—Place of residence, education status, occupation, wealth index, pregnancy, and religion were all believed to have an impact on women's time for premarital cohabitation.

### Statistical analysis

Descriptive analysis and shared frailty models were applied to achieve the main goal of the study and draw significant conclusions.

### Shared frailty models

The survival time, which is conditional on the random term called frailty ($u_i$, in cluster $i$ ($1 < i < n$)) are assumed to be independent and the proportional hazard model assumes

$$h_{ij}(t/X_{ij}, u_i) = \exp(\beta' X_{ij} + u_i) h_0(t) \tag{1}$$

Whereas an alternative to the proportional hazards assumption does not hold, the accelerated failure time frailty model assumes

$$h_{ij}(t/X_{ij}, u_i) = \exp(\beta' X_{ij} + u_i) h_0(\exp(\beta' X_{ij} + u_i) t \tag{2}$$

Where $i$ indicates the $i^{th}$ cluster, $j$ indicates the $j^{th}$ individual in the $i^{th}$ cluster, $h_0(t)$ is the baseline hazard, $u_i$ is the random term for all subjects in cluster $i$, $X_{ij}$ is the vector of covariates for subject $j$ in cluster $i$, and $\beta$ is the vector of regression coefficients. We assumed that $Z$, *where $Z$ = exp($u_i$)* has the gamma or inverse Gaussian distribution so that the hazard function depends upon this frailty that acts multiplicatively on it. The main assumption of a shared frailty model is that all individuals in cluster $i$ share the same value of frailty $Z_i$ ($i = 1,2,...,n$), that is why the model is called the shared frailty model. The survival time is assumed to be conditionally independent with respect to the shared (common survival times) frailty. This shared frailty is the cause of the dependence between survival times within the clusters. To study the effect of the

candidate covariates on women's time to premarital cohabitation, a first univariable analysis was conducted, with each candidate covariate being fitted into its model. The multivariable analysis included covariates that were found to be significant in the univariable analysis. The study used the Exponential, Weibull, and Log-logistic distributions for the baseline hazard function, as well as the Gamma and Inverse-Gaussian frailty distributions for the multivariable survival analysis. It was done with all significant factors from the univariable analysis.

## Results

### Descriptive analysis

The study included a total of 15,683 women from nine regional states and two city administrations. This study was interested in the time interval between the woman's birth date and the time of cohabitation. About 11,405 (72.7%) of study participants had experienced the occurrence (cohabited). Premarital cohabitation occurred at a median age of 18 years, with a minimum and maximum observed event time of 10 and 43 years, respectively. Premarital cohabitation was most common among women under the age of 15 years, with 1653 (10.5 percent) of them (Table 1).

**Table 1. Descriptive summary of time to premarital cohabitation of Ethiopian women.**

| Covariates | Categories | Status | | Total (%) |
|---|---|---|---|---|
| | | Censored (%) | Event (%) | |
| Education | No education | 527 (3.4%) | 6506 (41.5%) | 7033 (44.8%) |
| | Primary | 2004 (12.8%) | 3209 (20.5%) | 5213 (33.2%) |
| | Secondary | 1181 (7.5%) | 1057 (6.7%) | 2238 (14.3%) |
| | Higher | 566 (3.6%) | 633 (4.0%) | 1199 (7.6%) |
| Occupation | No | 2722 (17.4%) | 7289 (46.5%) | 10011 (63.8%) |
| | Yes | 1556 (9.9%) | 4116 (26.2%) | 5672 (36.2%) |
| Pregnancy | No | 4269 (27.2%) | 10292 (65.6%) | 14561 (92.8%) |
| | Yes | 9 (0.1%) | 1113 (7.1%) | 1122 (7.2%) |
| Wealth index | Poor | 981 (6.3%) | 4959 (31.6%) | 5940 (37.9%) |
| | Middle | 470 (3.0%) | 1532 (9.8%) | 2002 (12.8%) |
| | Rich | 2827 (18.0%) | 4914 (31.3%) | 7741 (49.4%) |
| Religion | Orthodox | 2047 (13.1%) | 4366 (27.8%) | 6413 (40.9%) |
| | Catholic | 25 (0.2%) | 66 (0.4%) | 91 (0.6%) |
| | Protestant | 823 (5.2%) | 1991 (12.7%) | 2814 (17.9%) |
| | Muslim | 1364 (8.7%) | 4845 (30.9%) | 6209 (39.6%) |
| | Others | 19 (0.1%) | 137 (0.9%) | 156 (1.0%) |
| Place of residence | Urban | 2165 (13.8%) | 3183 (20.3%) | 5348 (34.1%) |
| | Rural | 2113 (13.5%) | 8222 (52.4%) | 10335 (65.9%) |
| Region | Tigray | 443 (2.8%) | 1239 (7.9%) | 1682 (10.7%) |
| | Afar | 166 (1.1%) | 962 (6.1%) | 1128 (7.2%) |
| | Amhara | 372 (2.4%) | 1347 (8.6%) | 1719 (11.0%) |
| | Oromia | 430 (2.7%) | 1462 (9.3%) | 1892 (12.1%) |
| | Somali | 302 (1.9%) | 1089 (6.9%) | 1391 (8.9%) |
| | Benishangul G. | 240 (1.5%) | 886 (5.6%) | 1126 (7.2%) |
| | SNNPR | 556 (3.5%) | 1293 (8.2%) | 1849 (11.8%) |
| | Gambela | 206 (1.3%) | 829 (5.3%) | 1035 (6.6%) |
| | Harari | 227 (1.4%) | 679 (4.3%) | 906 (5.8%) |
| | Addis Ababa | 957 (6.1%) | 867 (5.5%) | 1824 (11.6%) |
| | Dire Dawa | 379 (2.4%) | 752 (4.8%) | 1131 (7.2%) |

**Table 2. AIC values of the parametric shared frailty models.**

| Model | | AIC |
|---|---|---|
| **Baseline hazard function** | **Frailty Distribution** | |
| Exponential | Gamma | 94829.31 |
| | Inverse-Gaussian | 94830.34 |
| Weibull | Gamma | 68763.75 |
| | Inverse-Gaussian | 68764.41 |
| Log-logistic | Gamma | 65144.36 |
| | Inverse-Gaussian | 65145.87 |

*AIC = Akaike's Information Criteria.*

## Multivariable analysis of shared frailty models

In this study, the AIC value of the Log-logistic-Gamma shared frailty model (65144.36) was found to be the lowest of all shared frailty models, indicating that the Log-logistic-Gamma shared frailty model is the most efficient model for describing time-to-premarital cohabitation data (Table 2).

Women's education, occupation, pregnancy, and place of residence were significant covariates of premarital cohabitation, according to an analysis based on a Log-logistic-Gamma shared frailty model; however, wealth index and religion were not. The women's survival duration of premarital cohabitation was shortened by a factor of ($\phi$ = 0.714, 0.455, and 0.385) for primary, secondary, and higher education, respectively, at a 5% level of significance. In other words, women who have completed at least primary school have a shorter premarital cohabitation survival period than the reference group. For pregnant women, premarital cohabitation survival time was shortened by a factor of ($\phi$ = 0.826), at a 5% level of significance. An acceleration factor larger than one suggests that premarital cohabitation is being prolonged.

Accordingly, women who had occupation ($\phi$ = 1.095) and women of rural Ethiopia ($\phi$ = 1.138) have prolonged time to premarital cohabitation when compared to their corresponding reference categories. In other words, they have a higher expected survival time than their corresponding reference categories. The frailty in this model is assumed to follow a Gamma distribution with a mean of 1 and a variance equal to theta ($\theta$). The heterogeneity in the population of the region that was used as a cluster was estimated by the selected model is $\theta$ = 0.524 and the dependence within the clusters (region) is measured by Kendall's tau at $\tau$ = 0.207 (20.7%). A variance of zero ($\theta$ = 0) would indicate that the frailty component does not contribute to the model. A likelihood ratio test for the hypothesis $\theta$ = 0 is shown at the bottom of Table 3 and indicates that a chi-square value of 434.66 with one degree of freedom resulted in a highly significant $p$-value <0.001. This implied that the frailty component had a significant contribution to the model. The estimate of the shape parameter in the Log-Logistic-Gamma shared frailty model is $\gamma$ = 8.405. This value shows the shape of the hazard function is unimodal because the value is greater than unity, i.e., it increases up to some time and then decreases (Table 3).

## Model diagnostics

### Diagnostic plots of the parametric baselines

### i. Diagnostic plot of the Log-logistic distribution

To check the adequacy of the baseline hazard, the log-logistic distribution is plotted by

$log\left(\frac{\hat{S}(t)}{1-\hat{S}(t)}\right)$ with the logarithm of time of the study. If the plot is linear, the given baseline

**Table 3. Multivariable analysis using Log-logistic-Gamma frailty model.**

| Covariates | Estimate | SE | *p* value | ϕ | 95% CI for ϕ |
|---|---|---|---|---|---|
| Education | | | | | |
| No education (Ref.) | 1 | | | | |
| Primary | -0.337 | 0.023 | <0.001 *** | 0.714 | [0.669, 0.759] |
| Secondary | -0.788 | 0.037 | <0.001 *** | 0.455 | [0.382, 0.527] |
| Higher | -0.954 | 0.047 | <0.001 *** | 0.385 | [0.293, 0.477] |
| Occupation | | | | | |
| No (Ref.) | 1 | | | | |
| Yes | 0.091 | 0.021 | <0.001 *** | 1.095 | [1.054, 1.136] |
| Wealth index | | | | | |
| Poor (Ref.) | 1 | | | | |
| Middle | -0.013 | 0.031 | 0.677 | 0.987 | [0.926, 1.048] |
| Rich | -0.017 | 0.028 | 0.549 | 0.983 | [0.928, 1.038] |
| Pregnancy | | | | | |
| No (Ref.) | 1 | | | | |
| Yes | -0.191 | 0.032 | <0.001 *** | 0.826 | [0.763, 0.889] |
| Religion | | | | | |
| Orthodox (Ref.) | 1 | | | | |
| Catholic | 0.017 | 0.125 | 0.890 | 1.017 | [0.772, 1.262] |
| Protestant | -0.009 | 0.035 | 0.803 | 0.991 | [0.922, 1.060] |
| Muslim | 0.030 | 0.029 | 0.302 | 1.030 | [0.974, 1.087] |
| Others | -0.094 | 0.090 | 0.296 | 0.910 | [0.734, 1.087] |
| Place of residence | | | | | |
| Urban (Ref.) | 1 | | | | |
| Rural | 0.129 | 0.033 | <0.001 *** | 1.138 | [1.073, 1.202] |

Source: EDHS 2016, Ethiopian Demographic and Health Survey Data 2016.

*significant at 5% level, θ = Variance random effect, τ = Kendall's Tau, λ = scale, γ = shape, Ref. = a reference category, SE = standard error of estimate, ϕ = Accelerated factor, CI = Confidence interval.

τ = 0.207 γ = 8.405 λ = 0.112 Theta(θ) = 0.524 (SE = 0.026).

Likelihood-ratio test of theta = 0: Chi-square = 434.66

P-value<0.001***.

distribution is appropriate for this dataset. The plot of log-logistic distribution is more than other plots (Fig 1).

## ii. Diagnostic plot of the Exponential distribution

The diagnostic plot of the exponential baseline distribution is plotted by the $-log(S(t))$ with the time of the study to check its adequacy. The plot of the exponential baseline distribution (Fig 2) is not linear as compared to the plot of the log-logistic baseline distribution (Fig 1).

## iii. Diagnostic plot of the Weibull distribution

To check the adequacy of the baseline hazard, the Weibull is plotted by $log(-log(S(t))$ with the logarithm of the time of the study. The plot of Weibull baseline distribution (Fig 3) is not linear as compared to the plot of the log-logistic baseline distribution (Fig 1).

## loglogistic

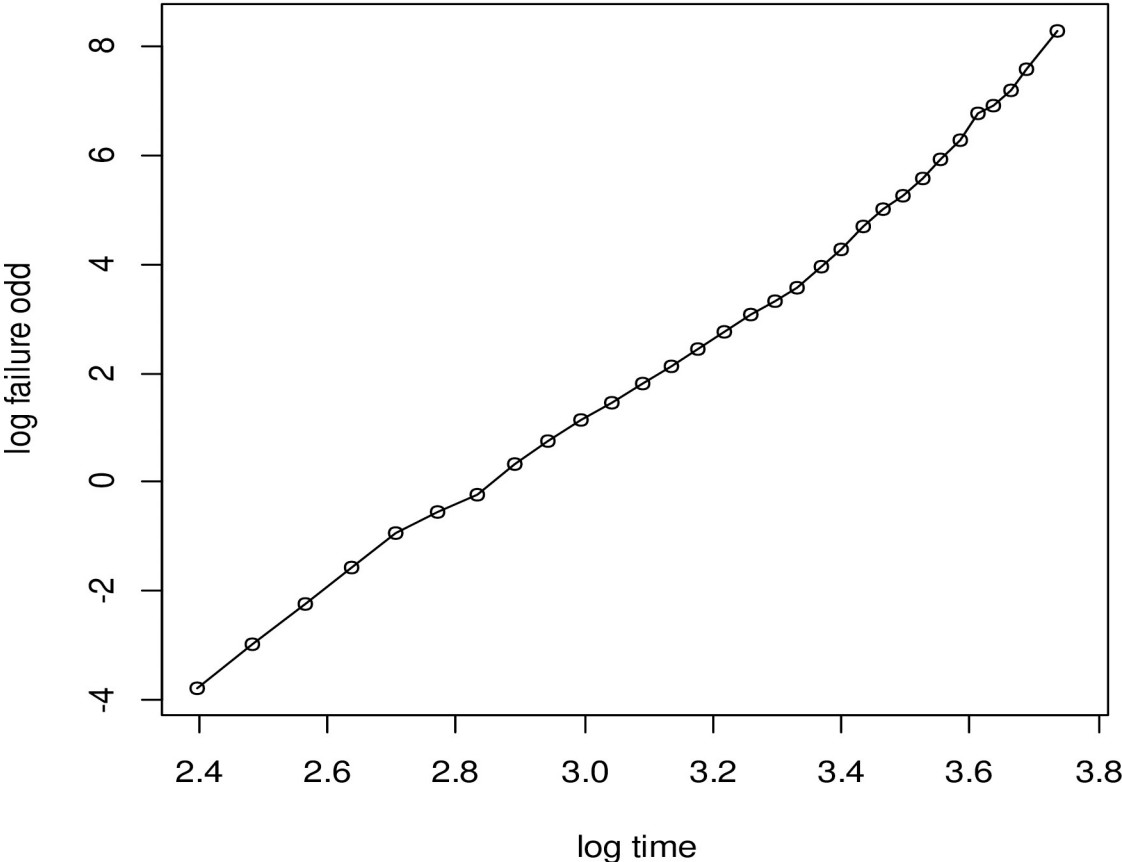

**Fig 1. Graph of Log-logistic baseline distribution to evaluate the assumption of linearity.**

## Discussion

The current study aimed to identify the determinants of time-to-premarital cohabitation among women in Ethiopia using some parametric shared frailty models. According to this study, premarital cohabitation (an event) was found to be 72.7%. This proportion is higher compared to the findings from the USA [5], Nigeria [6], and East Africa [7], where 70%, 65.2%, and 44% of women cohabited before marriage, respectively. The possible reason for this higher figure might be due to the variations in sociocultural and religious values and beliefs.

The cohabitation time of the women starts from their tenth year of birth to their forty-third year. To identify the determinants of time to premarital cohabitation, the AIC criteria were used to compare the model distributions, with the model with the lowest AIC being considered the best [21]. The most appropriate model to represent the time-to-premarital cohabitation data was the Log-logistic-Gamma frailty model, which had an AIC value of 65144.36. Using data from the EDHS 2016, the region was employed as frailty (clustering effect) to model the predictors of time to premarital cohabitation among Ethiopian women. For the Log-logistic-Gamma shared frailty model, the clustering effect was significant (p-value < 0.001). This confirmed regional heterogeneity by assuming that women in the same region have identical risk

# exponential

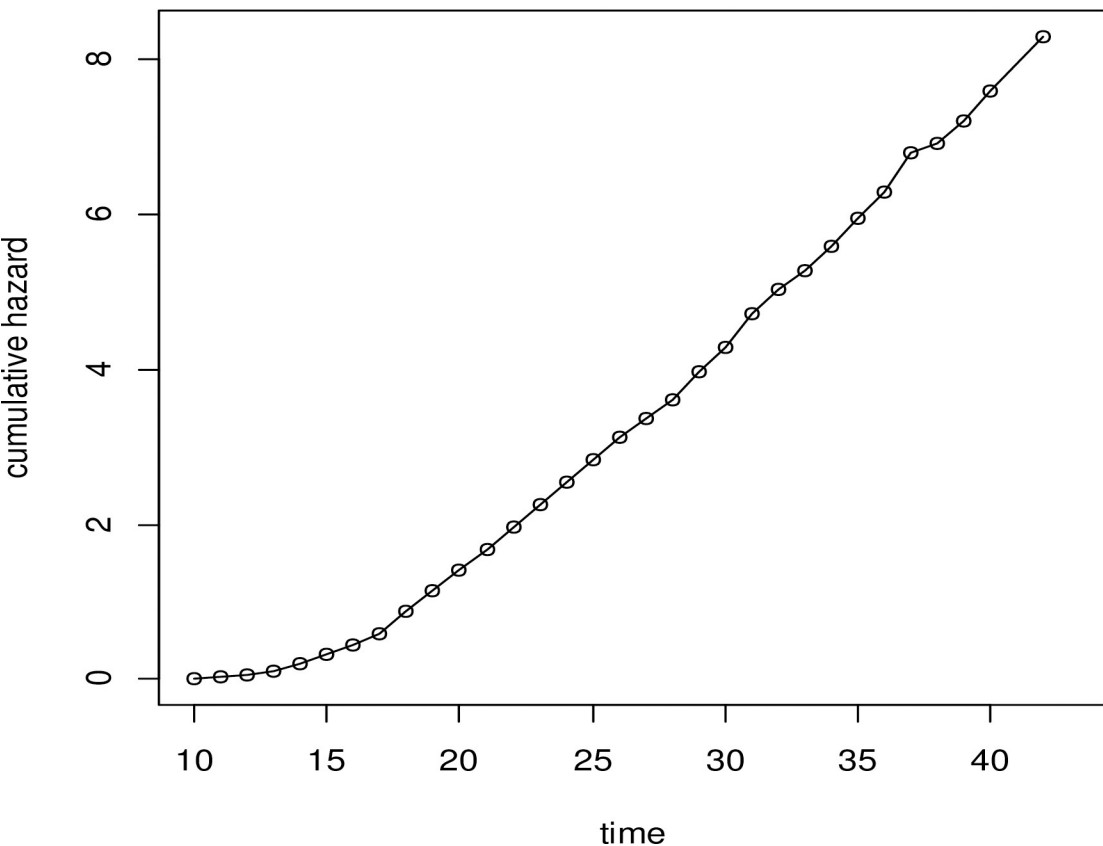

**Fig 2. Graph of Exponential baseline distribution to evaluate the assumption of linearity.**

factors for premarital cohabitation. That is, the correlation within regions cannot be ignored, and the clustering effect was crucial in hazard function modeling. Graphs were used to verify the appropriateness of the baseline distribution (Figs 1–3). The baseline Log-logistic plot was more linear than the other plots in the baseline distribution plots. This indicates that the log-logistic distribution is appropriate for the time-to-premarital cohabitation data. This result was compatible with [22–24].

According to the model prediction, place of residence significantly predicts time to premarital cohabitation among Ethiopian women. Women residing in the country's rural areas had a better chance of surviving premarital cohabitation than women who resided in urban areas. A similar study that has been conducted in SSA [4] revealed that urban women were more likely to cohabit compared to their rural counterparts. As urban areas are densely populated, there is limited access to occupations and accommodations, which can cause women to cohabit. In addition, the association might be due to rural communities' ingrained moral and cultural values that prohibit living together before marriage.

A significant association was found between women's educational status and time-to-premarital cohabitation. Women who attended primary school, secondary school, and higher education had significantly shorter time-to-premarital cohabitation compared to women who were not educated. This association is consistent with the studies conducted in the USA, Spain, Italy, SSA, and East Africa [4,25–27] which revealed that women who attended at least

## weibull

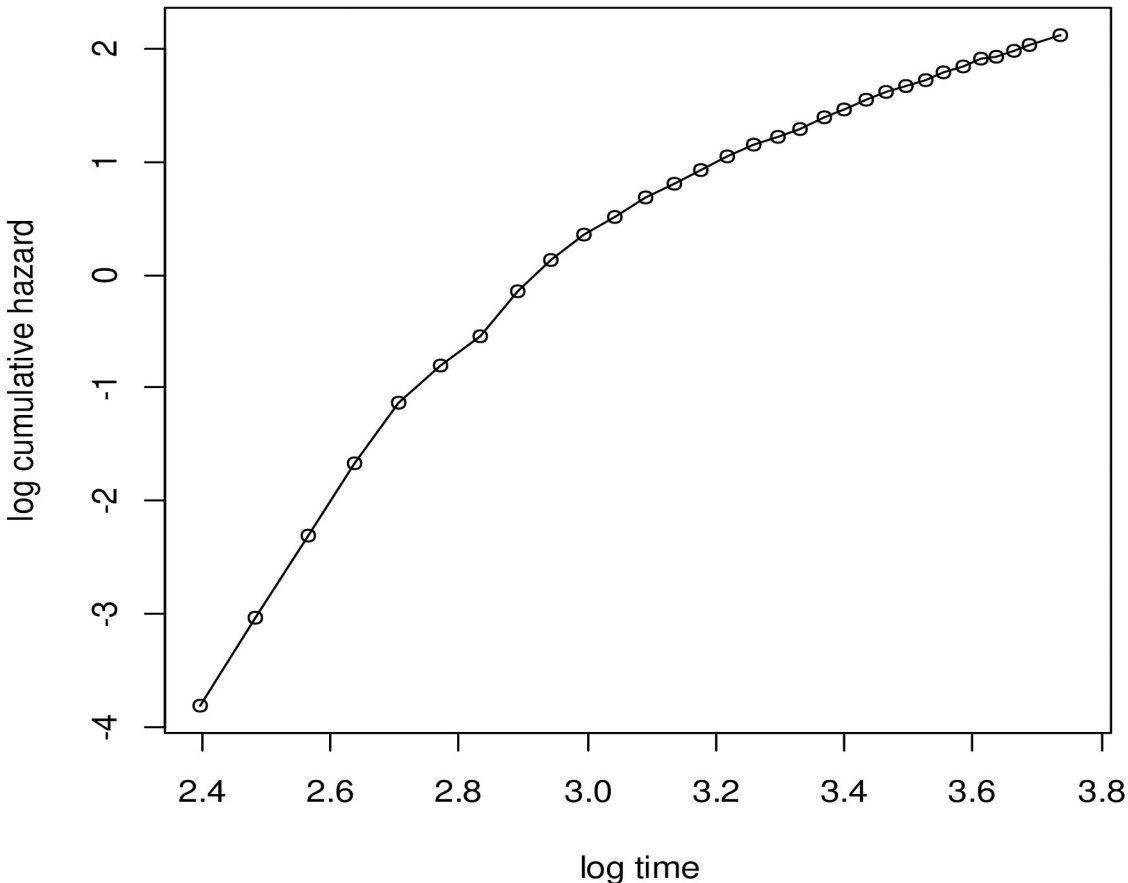

**Fig 3. Graph of Weibull baseline distribution to evaluate the assumption of linearity.**

primary school were more exposed to premarital cohabitation than those who were uneducated. Factually, a girl who goes outdoors to study has a higher probability of practicing premarital cohabitation than their counterparts, as loneliness is a cause of premarital cohabitation.

The results of this study also revealed that occupation was a significant predictor of time-to-premarital cohabitation among Ethiopian women. This shows that women who had an occupation had a longer survival time for premarital cohabitation compared to their counterparts. This finding is supported by the previous studies [25,27] where women who had an occupation were less vulnerable to premarital cohabitation. We thought this was a plausible association because a lack of occupation, the very source of accommodation income, can help women become independent before marriage.

A significant association was also found between pregnancy and time-to-premarital cohabitation. Women who were pregnant had significantly shortened time-to-premarital cohabitation as compared to non-pregnant women. This finding is in line with study findings from Spain [25], which found that pregnant women were more prone to premarital cohabitation than non-pregnant women. This relates to the certainty that a man and woman decide to live

together once the woman gets pregnant, which happens earlier in marriage, aiming to take care of the pregnancy.

The strength of the current study is its utilization of national data and a strong model that could guarantee statistical estimations. However, this study could have certain shortcomings. Since the information collected was self-reported, it would overestimate or underestimate the phenomenon. As the study utilized secondary data, other potential variables that could influence the time to premarital cohabitation and clustering effects would be missed in the current study. Hence, future researchers need to consider a primary, longitudinal study in the area. This study did not explore possible qualitative attributes of premarital cohabitation. As premarital cohabitation is a social problem, it might be entrenched in different socio-cultural and psychosocial beliefs that should be explored.

## Conclusions

The proportion of premarital cohabitation among Ethiopian women is higher than the proportion in the SSA and East Africa. Their cohabitation time was between the tenth and forty-third birth dates of women. According to the Log-logistic-Gamma shared frailty model, place of residence, occupation, educational status, and being pregnant were found to be factors determining the time to premarital cohabitation.

## Implications for practice

Premarital cohabitation is rising globally and among Ethiopian women. Hence, bearing in mind the negative effects of premarital cohabitation on the health, later marital relations, and psychology of couples, as well as its harm to the health and psychology of their children, the concerned bodies need to endorse intervention strategies in the area, including educating or disseminating information about the hazards of premarital cohabitation. Also, the concerned bodies should consider the identified predictor of premarital cohabitation when setting intervention strategies.

## Supporting information

**S1 File. A STROBE statement checklist.**
(DOCX)

**S1 Data. Data of our manuscript in csv (Comma delimited) format.**
(CSV)

## Acknowledgments

We acknowledge the Ethiopian Central Statistical Agency (CSA) for providing us with the data.

## Author Contributions

**Conceptualization:** Woldemariam Erkalo Gobena, Mesfin Esayas Lelisho.

**Data curation:** Woldemariam Erkalo Gobena, Wubishet Gezimu, Gizachew Gobebo Mekebo.

**Formal analysis:** Woldemariam Erkalo Gobena, Wubishet Gezimu, Gizachew Gobebo Mekebo, Teramaj Wongel Wotale, Mesfin Esayas Lelisho.

**Investigation:** Woldemariam Erkalo Gobena, Wubishet Gezimu, Gizachew Gobebo Mekebo.

**Methodology:** Woldemariam Erkalo Gobena, Gizachew Gobebo Mekebo, Teramaj Wongel Wotale, Mesfin Esayas Lelisho.

**Project administration:** Woldemariam Erkalo Gobena, Wubishet Gezimu, Gizachew Gobebo Mekebo.

**Resources:** Woldemariam Erkalo Gobena, Gizachew Gobebo Mekebo.

**Software:** Woldemariam Erkalo Gobena, Gizachew Gobebo Mekebo, Teramaj Wongel Wotale.

**Supervision:** Woldemariam Erkalo Gobena, Wubishet Gezimu, Gizachew Gobebo Mekebo, Teramaj Wongel Wotale.

**Validation:** Woldemariam Erkalo Gobena, Gizachew Gobebo Mekebo.

**Visualization:** Woldemariam Erkalo Gobena, Wubishet Gezimu, Gizachew Gobebo Mekebo, Teramaj Wongel Wotale.

**Writing – original draft:** Woldemariam Erkalo Gobena, Gizachew Gobebo Mekebo, Teramaj Wongel Wotale.

**Writing – review & editing:** Woldemariam Erkalo Gobena, Wubishet Gezimu, Gizachew Gobebo Mekebo, Teramaj Wongel Wotale, Mesfin Esayas Lelisho.

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
