## [Decision Letter · Decision Letter 0]

22 Jan 2024

PONE-D-23-34742Modeling the Determinants of Time-to-Premarital Cohabitation among Women in Ethiopia: A Comparison of Various Parametric Shared Frailty ModelsPLOS ONE

Dear Dr. GOBENA,

Thank you for submitting your manuscript to PLOS ONE. After careful consideration, we feel that it has merit but does not fully meet PLOS ONE’s publication criteria as it currently stands. Therefore, we invite you to submit a revised version of the manuscript that addresses the points raised during the review process.

We look forward to receiving your revised manuscript.

Kind regards,

Frank T. Spradley

Academic Editor

PLOS ONE

3. In the online submission form, you indicated that [The data used in current analysis will be available from corresponding author on reasonable request]. 

Reviewers' comments:

Reviewer's Responses to Questions

**Comments to the Author**

1. Is the manuscript technically sound, and do the data support the conclusions?

Reviewer #1: No

Reviewer #2: Yes

Reviewer #3: No

2. Has the statistical analysis been performed appropriately and rigorously? 

Reviewer #1: No

Reviewer #2: Yes

Reviewer #3: No

3. Have the authors made all data underlying the findings in their manuscript fully available?

Reviewer #1: Yes

Reviewer #2: Yes

Reviewer #3: Yes

4. Is the manuscript presented in an intelligible fashion and written in standard English?

Reviewer #1: Yes

Reviewer #2: No

Reviewer #3: No

5. Review Comments to the Author

Reviewer #1: The writing of the paper is not scientific enough. I recommend the authors to revise the entire paper.

• The figures are not clear at all for the reviewer to understand the functionality of the circuits. All figures must be replaced with new ones with better clarity.

• The proposed cell is not investigated in detail and the reviewer cannot follow the novelty.

• The simulations carried out are not sufficient at all. Different parameters must be included in the simulations by the authors.

• The paper has lack of enough novelty, contribution, and significant results for being published.

• The authors should rewrite the paper and include recent and state-of-the-art designs in simulations and include statements WHAT is the significant superiority of this paper in comparison to many other papers in the literature.

• There is a lack of figure of merit in comparison sections which should be considered when it comes to the novelty and superiority of this paper.

Reviewer #2: What is your motivation to study this area?

why not include other most important variables? such as age,...

why you select parametric shared frailty models? Why not use Cox PH, AFT,..

your conclusions and discussions quite poor. So that, revise it again

Reviewer #3: Thank you for the opportunity to review the manuscript “Modeling the Determinants of Time-to-Premarital Cohabitation among Women in Ethiopia: A Comparison of Various Parametric Shared Frailty Models”.

0.Title

“A Comparison of Various Parametric Shared Frailty Models”, is the main objective of the manuscript to compare different models?

1. Abstract

The abstract was not written in good manuscript format. Therefore, the author suggested writing the abstract in a well-understandable fashion by including an introduction, methods, result, and conclusion. Because the abstract is the mine version of your manuscript.

2. Introduction

The author stated that “As a result, cohabitation may alter people's perceptions of marriage, making themless dedicated and devoted to the institution” is that the authors opinion?

The authors also stated that “However, in Sub-Saharan Africa (SSA), where just

3 a few studies have been undertaken, there is a dearth of such literature”, the dearth of the literatures is may be due to the less important nature of this type of research question in SSA. More specifically, in Ethiopia, cohabitation is not a big problem due to strong cultural and religious protections. The authors also strengthen this idea by stating, “Premarital cohabitation was

frowned upon in Ethiopia and was regarded as a socially unacceptable practice.”. Therefore, cohabitation may not be a big deal.

“Previously conducted studies used logistic regression to identify factors of premarital cohabitation. However, logistic regression does not take into account censoring observations; hence, it is not applicable to time-to-event data. As a result, this study intends to fill a gap in the literature by utilizing parametric shared frailty models to investigate predictors of time to premarital cohabitation in Ethiopia”. For this scenario, logistic regression is more appropriate than survival analysis due to the nature of the premarital cohabitation, unless the objective of the study is methodological review.

In the last statement of the introduction of the manuscript, the authors stated that “Finally, this research is critical in the development of effective policies to educate women about the dangers of premarital cohabitation on their health and marriage.” Does premarital cohabitation only affect women's lives?

Note that the introduction of the manuscript is not well written; it should provide a clear and concise overview of the research, establish the context, highlight the significance of the study, and outline the research question or objectives.

3. Methods

Data Source: since the data used for the analysis is extracted from EDHS and is online, everyone can access it. Why did the authors retrieve the information from CSA?

Study population and variables: the authors stated that “in the families of selected clusters (regions), a total of 15,683 women between the ages of 10 and 43 years were identified.”. The age of the study population is not related to the research question; most often, premarital cohabitation is practiced in the youngest age groups (18-24). Studying premarital cohabitation among 10-year-old children is not even ethical. Therefore, the authors did not use appropriate inclusion criteria to select study populations. Furthermore, is the study population only women? Why not men?

The response variable definition was also incorrect: "time to premarital cohabitation." The authors should clearly explain the event, censored, and time definitions for the study. The covariates included in this study were also not supported by the literature (not explained in the introduction section of the manuscript). Most of the covariates included in the study were not potential factors for the response variable.

Methods of Data Analysis: Shared Frailty Models

The authors start the analysis of the frailty models without any motivation. It is normal to start analyzing the survival data from the non-parametric model, and then by checking each assumption of the model, you can extend the parametric model. The authors did not show any evidence to check whether there is clustering effect or not in any part of the manuscript. The authors also did not try to list any graphical presentations to explore the nature of the data, which helps get insight to select an appropriate model that better fit the data. especially for survival analysis.

4. Results:

The authors stated that “The study included a total of 15,683 women from nine regional states and two city administrations”, currently the region of Ethiopia is not only nine, and this old data leads to biased results.

The authors also stated that “this study was interested in the time interval between the women's birth date and the time of cohabitation.” What is the importance of studying this interval?

The authors reported that more than 72% of the respondents had experienced the event. Which is very exaggerated and unreliable. The minimum and maximum observed event time are reported as 10 and 43 years, with a median age of 18. Reporting that 10 year-old children have been experienced with cohabitation is unethical.

In multivariate analysis the authors report different model with the corresponding AIC. Basically, the variables included in the model were not supported by the literatures, and the model was fitted without assessing the assumptions of the models. Therefore, the result presented may not be reliable and biased. Furthermore, selecting the better model based on the AIC is not reliable because it is affected by the sample size.

6. PLOS authors have the option to publish the peer review history of their article (what does this mean?). If published, this will include your full peer review and any attached files.

Reviewer #1: No

Reviewer #2: No

Reviewer #3: **Yes: **Gebrekidan Ewnetu Tarekegn

---

## [Author Response · Author response to Decision Letter 0]

2 Mar 2024

Reviewer #1: 

Dear reviewer, thank you for reviewing and forwarding such important comments that enhance the readability and quality of our manuscript. For easy check-up, our responses are highlighted in turquoise, and all the changes in the manuscript are highlighted in yellow. 

1. The writing of the paper is not scientific enough. I recommend the authors revise the entire paper. 

 Authors’ Response: Thanks you very much. We have revised the entire paper.

2. The figures are not clear enough for the reviewer to understand the functionality of the circuits. All figures must be replaced with new ones with better clarity. 

Authors’ Response: Thank you very much. We have revised and formatted figures as per your comment and the journal’s figure formatting guideline. 

3. The proposed cell is not investigated in detail, and the reviewer cannot follow the novelty.

Authors’ Response: Thank you a lot. We thought the study met its main objectives and kept its novelty as justified in the introduction section. If you have a specific point comment or suggestion on this issue, please, feel free to depict it and we will fix it. 

4. The simulations carried out are not sufficient at all. Different parameters must be included in the simulations by the authors. 

Authors’ Response: Thank you for your nice comment. We have used more than eight parameters (more than six parameters for the factors, scale parameter, and shape parameter) in the simulation. We believe that those parameters are enough for the simulation. Or, would you suggest the possible parameters to be added to the simulation, please? If so, we will at least mention it as a limitation and suggest future researchers to consider. 

5. The paper lacks novelty, contribution, and significant results to be published. 

Authors’ Response: Thank you so much. To the level of our literature search, few studies have been conducted on premarital cohabitation using logistic regression that cannot consider censoring observations. Given this, uniquely, the current study determines predictors of time-to-premarital cohabitation among Ethiopian women using parametric shared frailty models. We hope the findings of this study will benefit policymakers in setting out interventions regarding the mitigation of premarital cohabitations. Moreover, the study may benefit sociologists and clinical researchers. To increase its scientific quality and readability, we have revised the whole manuscript.

6. The authors should rewrite the paper, include recent and state-of-the-art designs in simulations, and include statements about the significant superiority of this paper in comparison to many other papers in the literature. 

Authors’ Response: Thank you so much for your insightful suggestion. We have revised our manuscript based on your suggestions as highlighted in the revised manuscript.

7. There is a lack of merit in the comparison sections, which should be considered when it comes to the novelty and superiority of this paper. 

Authors’ Response: Thank very much for your crucial suggestion. We have revised our manuscript as per your suggestion. 

Reviewer #2: 

Dear reviewer, we are so grateful for your very crucial and constructive comments that help to improve the quality of the manuscript. Your comments are so logical and very important. For easy check-up, our responses are highlighted in turquoise, and all the changes in the manuscript are highlighted in yellow. 

1. What is your motivation to study this area? 

Authors’ Response: Thank you a lot. A few studies have been conducted on factors that cause premarital cohabitation. No studies have been conducted previously on time to premarital cohabitation or age at premarital cohabitation using survival analysis. Survival analysis focuses on the analysis of time-to-event data, where the primary interest is in the time until an event of interest occurs. This type of analysis is commonly used in medical research, engineering, economics, and other fields to study the time until a particular event occurs. Survival analysis techniques account for censored data, where the event of interest has not occurred for some individuals by the end of the study or follow-up period. 

2. Why not include the other most important variables? such as age,... 

 Authors’ Response: Thank for you insightful suggestion. We have used age as a response variable, i.e., time to premarital cohabitation. It is calculated as the time between birth and the age of first cohabitation. For this study, cohabited women were the focus of the study, and women who did not participate in the event (not cohabited) were censored. 

3. Why do you select parametric shared frailty models? Why not use Cox PH, AFT, etc.? 

Authors’ Response: We appreciate the reviewer for this crucial comment. We have selected parametric shared frailty models over Cox regression because they allow for the inclusion of random effects, which can account for unobserved heterogeneity within clusters or groups in the data. This is particularly useful when there is correlation among individuals within the same cluster, such as in family or geographical studies. Thus, we have used the national data, which is clustered by region. Parametric shared frailty models can handle time-to-event data with non-proportional hazards more effectively than Cox regression, as they allow for the estimation of baseline hazard functions and time-varying covariate effects. Overall, the use of parametric shared frailty models can lead to more accurate and reliable outcomes in certain scenarios, particularly when dealing with correlated or clustered survival data. 

4. Your conclusions and discussions are quite poor. So revise it again. 

Authors’ Response: Thank you very much for your intellectual suggestion. We have revised the discussions and conclusions as per your suggestion.

 Reviewer #3: 

Dear reviewer, thank you for reviewing and forwarding such important comments that enhance the readability and quality of our manuscript. For easy check-up, our responses are highlighted in turquoise, and all the changes in the manuscript are highlighted in yellow.

1. The title

“A Comparison of Various Parametric Shared Frailty Models” is the main objective of the manuscript: to compare different models? 

Authors’ Response: Thank you so much. “A Comparison of Various Parametric Shared Frailty Models” is not the main objective of the study. We have used parametric shared frailty models to determine factors of time to premarital cohabitation (or factors of age at premarital cohabitation). Since our data is clustered. For the clustered data, parametric shared fragility models are appropriate models to analyze time-to-event data, where the primary interest is in the time until an event of interest occurs. In our study, cohabiting women were an event of interest. Women who did not participate in the event (not cohabited) were censored.

2. Abstract 

The abstract was not written in good manuscript format. Therefore, the author suggested writing the abstract in a well-understandable fashion by including an introduction, methods, result, and conclusion. Because the abstract is the original version of your manuscript. 

Authors’ Response: Thank you so much for your insightful comment. We have re-written the abstract in a good manuscript format by classifying it into four sections: an introduction, methods, result, and conclusion. 

3. Introduction: 

The author stated that “as a result, cohabitation may alter people's perceptions of marriage, making them less dedicated and devoted to the institution.” Is that the author's opinion? The authors also stated that “However, in Sub-Saharan Africa (SSA), where only a few studies have been undertaken, there is a dearth of such literature." The dearth of literature may be due to the less important nature of this type of research question in SSA. More specifically, in Ethiopia, cohabitation is not a big problem due to strong cultural and religious protections. The authors also strengthen this idea by stating, “Premarital cohabitation was frowned upon in Ethiopia and was regarded as a socially unacceptable practice.” Therefore, cohabitation may not be a big deal. 

Authors’ Response: Thank you for your comment. We think nobody is happy when his or her daughter or sister is cohabited or married at an early age. We have many different cultures and religions in Ethiopia. Our cultures are even different from one zonal administration to another in the same region. Some cultures allow premarital cohabitation and early marriage, and most do not. Even though premarital cohabitation is socially taboo, it's now more prevalent in Ethiopia. We have used national data and EDHS data. The data revealed that out of 15,683 women in the study, 72% of them cohabited. And also, some studies conducted previously revealed that premarital cohabitation has a negative impact on marriage. Cohabited couples are linked to a higher probability of divorce, and they exhibit inferior interpersonal skills and less predictable attitudes toward marriage when compared to married couples. Premarital cohabitation may alter people's perceptions of marriage, making them less dedicated and devoted to the institution. Furthermore, premarital cohabitation may result in unintended pregnancy and abortion. It’s known that unintended pregnancy and abortion adversely affect the health of women. Thus, it’s good to give attention to the event. Women should be given health education to be aware of the dangers of premarital cohabitation for their marriage and health. 

4. “Previously conducted studies used logistic regression to identify factors of premarital cohabitation. However, logistic regression does not consider censoring observations; hence, it is not applicable to time-to-event data. As a result, this study intends to fill a gap in the literature by utilizing parametric shared frailty models to investigate predictors of time to premarital cohabitation in Ethiopia”. For this scenario, logistic regression is more appropriate than survival analysis due to the nature of the premarital cohabitation, unless the objective of the study is a methodological review. 

Authors’ Response: Thank you so much for your comment. However, this study analyzes time-to-premarital cohabitation, i.e., the time between birth and the age at first cohabitation. For this study, cohabiting women were an event of interest. Women who did not participate in the event (not cohabited) were censored. Thus, logistic regression does not consider censoring observations; hence, it is not applicable to time-to-event data. 

5. In the last statement of the introduction of the manuscript, the authors stated that “Finally, this research is critical in the development of effective policies to educate women about the dangers of premarital cohabitation on their health and marriage.” Does premarital cohabitation only affect women's lives? 

 Authors’ Response: We appreciate the reviewer for this intellectual comment. Premarital cohabitation affects more than just women. It also affects men. But premarital cohabitation affects women more because it causes unintended pregnancy and abortion. So, that is why we were motivated to do this research on women only. 

6. Note that the introduction of the manuscript is not well written; it should provide a clear and concise overview of the research, establish the context, highlight the significance of the study, and outline the research question or objectives. 

Authors’ Response: Thank you so much. We have revised the introduction part.

Methods 

Data Source: since the data used for the analysis is extracted from EDHS and is online, everyone can access it. Why did the authors retrieve the information from CSA? 

Authors’ Response: The CSA of Ethiopia is responsible for EDHS data. To download the data that you want, you first have to register and submit information about the research you plan to conduct. Then, the system displays your data for download. 

Study population and variables: the authors stated that “in the families of selected clusters (regions), a total of 15,683 women between the ages of 10 and 43 years were identified.” The age of the study population is not related to the research question; most often, premarital cohabitation is practiced in the youngest age groups (18–24). Studying premarital cohabitation among 10-year-old children is not even ethical. Therefore, the authors did not use appropriate inclusion criteria to select study populations. Furthermore, is the study population only women? Why not men? 

Authors’ Response: Thank you very much for your comment. It’s good to consider inclusion and exclusion criteria when selecting the study population. However, we have used secondary data (already existing national data and EDHS data). We thought that it wasn't necessary to consider inclusion and exclusion criteria for secondary data. The national data that we have used revealed that the minimum age of premarital cohabitation was 10 and the maximum was 43 years, and the median age was 18 years. We cannot distort the truth because the data is available online for all. Anyone can access it. There are cultures in Ethiopia that allow girls to marry at the age of 11, 12, or 13. You can take the northern part of Ethiopia as a good example. So why is studying premarital cohabitation among 10-year-old girls unethical? It’s not unethical. We didn’t collect data by ourselves. It’s EDHS data (secondary data) available at https://www.dhsprogram.com/. 

The response variable definition was also incorrect: "time to premarital cohabitation." The authors should clearly explain the event, censored, and time definitions for the study. The covariates included in this study were also not supported by the literature (not explained in the introduction section of the manuscript). Most of the covariates included in the study were not potential factors for the response variable. 

Authors’ Response: Thank you for your nice comment. The response variable is the time to premarital cohabitation among women of reproductive age, which is measured in years. It is calculated as the time between birth and the age of first cohabitation. For this study, cohabited women were considered as the event, and women who did not participate in the event (not cohabited) were censored. 

Methods of Data Analysis/Shared Frailty Models: The authors start the analysis of the frailty models without any motivation. It is normal to start analyzing the survival data from the non-parametric model, and then, by checking each assumption of the model, you can extend the parametric model. The authors did not show any evidence to check whether there is a clustering effect or not in any part of the manuscript. The authors also did not try to list any graphical presentations to explore the nature of the data, which helps get insight into selecting an appropriate model that better fits the data. Especially for survival analysis. 

Authors’ Response: Due to the nature of the data, we have used parametric shared frailty models. The EDHS data are the clustered data. It’s clustered by enumeration units, kebeles, woredas, zones, and regions. In the analysis of the data, the frailty/clustering effect was significant (p value < 0.001). This confirmed regional heterogeneity by assuming that women in the same region have identical risk factors for premarital cohabitation. That is, the correlation within regions cannot be ignored, and the clustering effect was crucial in hazard function modeling. Graphs were used to verify the appropriateness of the baseline distribution (Fig. 1, Fig. 2, and Fig. 3). The baseline log-logistic plot was more linear than the other plots in the baseline distribution plots. We have used diagnostic plots of the parametric baselines to select an appropriate model that better fits the data. If the plot is linear, the given baseline distribution is appropriate for this dataset. The plot of log-logistics was more linear than other plots (Fig. 1). 

7. Results:

 The authors stated that “the study included a total of 15,683 women from nine regional states and two city administrations." Currently, currently the region of Ethiopia is not only nine, and this old data leads to biased results. The authors also stated that “this s

---

## [Decision Letter · Decision Letter 1]

22 Apr 2024

Modeling determinants of time-to-premarital cohabitation among Ethiopian women using parametric shared frailty models

PONE-D-23-34742R1

Dear Dr. GOBENA,

We’re pleased to inform you that your manuscript has been judged scientifically suitable for publication and will be formally accepted for publication once it meets all outstanding technical requirements.

Kind regards,

Faten Amer, PhD in Health Sciences

Academic Editor

PLOS ONE

Reviewers' comments:

Reviewer's Responses to Questions

**Comments to the Author**

1. If the authors have adequately addressed your comments raised in a previous round of review and you feel that this manuscript is now acceptable for publication, you may indicate that here to bypass the “Comments to the Author” section, enter your conflict of interest statement in the “Confidential to Editor” section, and submit your "Accept" recommendation.

Reviewer #1: All comments have been addressed

2. Is the manuscript technically sound, and do the data support the conclusions?

Reviewer #1: Yes

3. Has the statistical analysis been performed appropriately and rigorously? 

Reviewer #1: Yes

4. Have the authors made all data underlying the findings in their manuscript fully available?

Reviewer #1: Yes

5. Is the manuscript presented in an intelligible fashion and written in standard English?

Reviewer #1: Yes

6. Review Comments to the Author

Reviewer #1: Final proof reading required by professional english order.Revision done by author as per coments and as of now its ok.

7. PLOS authors have the option to publish the peer review history of their article (what does this mean?). If published, this will include your full peer review and any attached files.

Reviewer #1: **Yes: **Ok

---

## [Editor Report · Acceptance letter]

2 May 2024

PONE-D-23-34742R1 

PLOS ONE

Dear Dr. GOBENA, 

I'm pleased to inform you that your manuscript has been deemed suitable for publication in PLOS ONE. Congratulations! Your manuscript is now being handed over to our production team.

Kind regards, 

on behalf of

Dr. Faten Amer 

Academic Editor

PLOS ONE